# Investigating Decision Mechanisms of Statutory Stakeholders in Flood Risk Strategy Formation: A Computational Experiments Approach

**Ifigeneia Koutiva \*, Archontia Lykou, Chris Pantazis and Christos Makropoulos**

Department of Water Resources and Environmental Engineering, School of Civil Engineering, National Technical University of Athens, Zografou 11527, Greece; alykou@central.ntua.gr (A.L.); xpanta@gmail.com (C.P.); cmakro@mail.ntua.gr (C.M.)

\* Correspondence: ikoutiva@mail.ntua.gr

**Abstract:** Cities at risk of extreme hydro-meteorological events need to be prepared to decrease the extent of the impacts. However, sometimes, authorities only react to catastrophes failing to proactively prepare against extremes. This can be a result of both absent structural protection measures and problematic governance. While for the first, models exist that can simulate the effect, the effect of the latter is difficult to quantify. This work aims to explore the effects that typical authorities' behaviour has on the decisions for preparing and protecting a city against floods. This behaviour includes how the different authorities decide, for example, on whether or not to cooperate with each other, build something, assign funding to something, etc. These decisions affect directly the preparedness against and the protection from flood events. For that matter, the institutional analysis framework was used to conceptualise the decision-making processes of authorities responsible for flood risk management. Based on this, an agent-based modelling tool has been created, enabling the exploration of the system's behaviour under different scenarios. The tool is used as a case study of the responsible authorities for flood protection in the city of Rethymno on the island of Crete, Greece. The tool has a user-friendly interface enabling the end-users to explore the drivers of decision-making processes under different conditions.

**Keywords:** decision making support; flood risk management; agent based modelling; socio-hydrology

## 1. Introduction

Flood risk management is dependent on biophysical, structural, political, economic, social and regulatory conditions. These conditions shape the behaviour of institutions which are usually trying to make trade-offs between costs, politics and flood risk management. The exploration of this behaviour may reveal the shaping mechanisms and eventually increase flood risk management performance by removing barriers and introducing enablers. Providing an "experimental" approach to the decision makers enables a better understanding of the way the system may react to changes (structural or operational). In hydro-informatics, agent-based modelling (ABM) is gaining ground as means to experiment with the behaviour of the social component of the water cycle [1]. The main reason for the prevalence of ABM is its ability to address problems that concern emergence arising from interactions between a system's individual components and their environment [2]. This method is ideal for simulating the dynamic interaction between different components of the same complex system, i.e., the social component and the water system component [3].

ABM is a computational intelligence application that is based on agents which are "computer systems situated in some environment, capable of autonomous action in this environment in order

to meet its design objectives" [4]. Essentially, the ABM is a form of computational social science [5], where the complex system of behaviour is generated by the interaction of the simple components. In other words, an agent has a set of features as well as a set of rules that define its behaviour, so that it can respond to the environment and interact with other agents, altering their own state, that of the environment and even that of other agents [6].

ABM tools aimed at simulation of water management have been developed in the past. Some examples are the ABM developed to simulate flood incident management and implemented for the coastal town of Towyn in the United Kingdom [7], the ABM developed in the NeWater project to explore mechanisms of resilience in the Amudarya basin [8], and the coordination and management of water networks in Bali [9]. Recently, ABMs have been developed to explore institutional behaviour, such as [10] which aims to explore the dynamical evolution of flood risk and vulnerability including the effects of insurance mechanisms and [11] exploring the impact of community policies on the evolution of flood risk. Additionally, ABMs have been recently linked to flood models to explore human–flood interactions and evaluate flood risk management options [12,13], calculate dynamic exposure to floods [14], and support planning evacuation procedures [15].

This work presents a tool developed to enable the experimentation with the decision-making process of authorities responsible for a city's flood risk management. This research was included within the activities of the EU funded PEARL project [16], which was completed at the end of 2017. The produced ABM was part of an overall Toolbox that was created to support decision-makers regarding flood risk management [17]. The developed tool includes the simulation of the decision-making process of authorities responsible for a city's flood protection and preparation. The tool supports an "experimental" approach, such as the one proposed by adaptive water resources management [8]. This enables decision-makers to better understand the way the system may react to changes (structural or operational) while preparing against extreme hydro-meteorological events. Agent based modelling (ABM) was selected to create this tool, mainly due to its ability to address problems that concern emergence arising from interactions between a system's individual components and their environment [2], and therefore to help simulate the dynamic interaction between different components of the same complex system, i.e., the socio-economic component and the water system component [1,3,18–20].

## 2. Materials and Methods

### 2.1. Methodology

In this research, the institutional analysis framework (IAF) developed by Ostrom [21,22] was used as a roadmap in order to conceptualise the decision-making processes of authorities responsible for flood risk management (see Figure 1). Based on Ostrom's theory, in order to analyse institutions, we need to identify the external variables that comprise the biophysical conditions, the attributes of the community based on historical data and previously implemented actions and the rules of the system identifying who may implement actions and whom and how these actions affect [23,24].

Additionally, as seen in the action situations element of the IAF in Figure 1, in order to specify how the elements of the system are transformed into outcomes we need to identify the characteristics of the actors involved, the positions they hold, and the set of actions that the actors take. Finally, in order to identify the potential outcomes, it is necessary to define the available information and the control over the outcomes of an action as well as the payoff rules that assign costs and benefits to an action's outcomes [22,24].

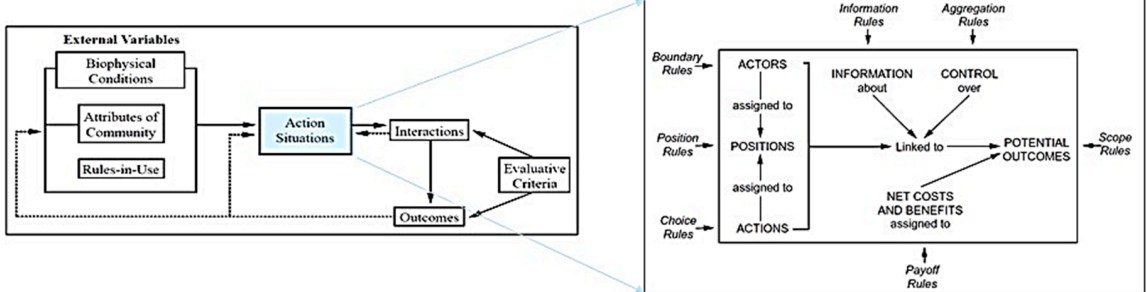

**Figure 1.** Institutional Analysis Framework and analysis of the action situations element (adapted from [22,24].

Figure 2 presents the implementation of the IAF to the flood protection decision making. The relevant biophysical conditions have been identified as the specific characteristics of the river basin, the climatic characteristics and the existing flood measures. The attributes of community include among others the historical flooding events and their effects to the area, including the decision makers' actions. In terms of the outcomes, these can be measured based on the performance of the area to protect, prevent and prepare against floods. Finally, the rules-in-use are something explicit in each case and need to be defined on a case by case basis. This representation was used to identify the main components of the decision-making process of flood risk management which was used in turn to design the developed ABM tool.

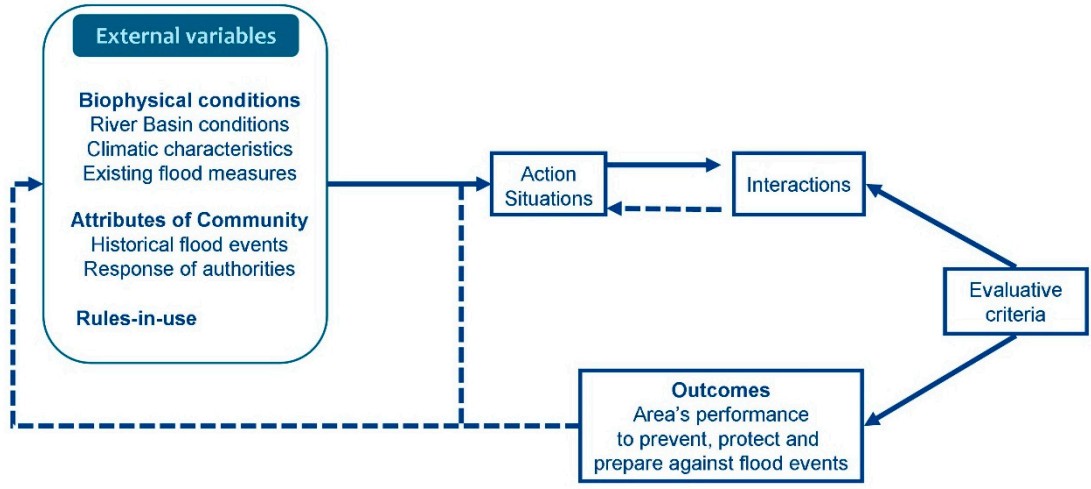

**Figure 2.** IAF of flood risk management decision making.

### 2.1.1. External Variables

The proposed ABM was based on the PEARL project's case study of the city of Rethymno in Crete Greece. Rethymno is situated at the Region of Crete in Greece and its population stands at 32,468 inhabitants according to Census 2011. Rethymno is the 3rd most populous urban area in the island of Crete with commercial, administrative, cultural and tourist activities along the north coast. The mean absolute altitude is about 15 m and the length along the coastline of the area under study is 8 km [25].

Rethymno is a city vulnerable to both rainfall and coastal floods. There have been many historic flood events which led to adverse effects for the city of Rethymno and eventually led to the implementation of engineering flood prevention and mitigation measures. Primarily cause of those flood events was heavy precipitation and insufficient flood protection infrastructure.

The city's special hydro-geomorphological conditions require high volumes of upstream water to pass through the city on the way to the sea. Furthermore, the historical city of Rethymno, was expanded without proper flood planning and streams delineation. On top of that, local businesses cover the drain grates of the storm water network to avoid unpleasant smells, which increases the flood water depth in the streets of the historical centre. Additionally, the coastal zone is exposed to strong N and NW winds, that overtop the port infrastructures, causing damage to both public and private properties.

In terms of flood risk management three distinct periods of time have been identified (as seen in Table 1) [26]. The first period from 1968 flood event until 1997 is characterized by urban floods with a high impact on properties and critical infrastructure assets mainly because of the lack of protection and prevention mechanisms. The second period, from 1998 to 2008, was characterised by the use of EU funds, and the political will of the then mayor Mr Archontakis, which made it possible for the construction of several flood protection engineering measures, like the Kamaraki and the Sinatsaki flood protection stream barriers. These construction measures were not supported by awareness raising and stakeholder engagement policies [26]. The third period is from 2009 up to this point and is characterised by the effects of the Greek economic crisis which inevitably decreased the available funding for flood risk management. Nevertheless, the existence of the flood infrastructure implemented in Rethymno puts the city in a far better position compared to other coastal cities in Greece.

**Table 1.** Evolution of flood risk management in Rethymno (adapted from [26]).

| Period | Characteristics |
|---|---|
| 1968–1997 | - lack of infrastructure<br>- major urban floods (1968, 1984, 1991)<br>- low funding opportunities |
| 1998–2008 | - last major urban flood (1999)<br>- large scale infrastructure<br>- EU funding |
| 2009 until today | - austerity measures decrease funds<br>- preferences for small-scale constructions<br>- stakeholder involvement |

2.1.2. Interactions

The action situations (Figure 1 right element) require the identification of the actors, their positions and the actions they may implement. As proposed in the Flood Directive 2007/60 [27], the responsible authorities for the protection of a city from flood events are the local authorities. These authorities may be divided for example depending on the spatial scale that they are responsible for, i.e., RBD or other relevant scale, etc. Relevant to our methodology are the responsibilities of authorities, as stated in the Flood Directive, for prevention, protection, and preparedness. Additionally, even though local authorities are responsible for actions, other stakeholders are usually also involved in the implementation of the actions [26]. This is a leverage point of the decision-making process, controlling the implementation of actions and thus making the cooperation between the local authorities and the stakeholders a focal point in our methodology.

In the case of Rethymno, a Learning and Action Alliance (LAA) [28] was created based on the sociogram of the stakeholders participating in flood risk management [29]. The sociogram identified the key stakeholders, their ways of interaction, and highlighted the gaps in communication and knowledge sharing. The main conclusion of this stakeholder analysis result was that there is a lack

of interaction among different stakeholders and barriers derived by different levels of hierarchy or conflicting interests [29].

The analysis of the action situations element is fed by the results from the two stakeholder workshops held in Rethymno where the LAAs participated. The local authorities of Rethymno (Municipality and Region) are identified as the main actors that are responsible for taking actions. However, more actors are involved in the decision making such as the water utility company and the general public.

During the 1st stakeholder workshop of Rethymno [30], the participants were asked to identify several aspects of the decision making process, such as who needs to cooperate with whom and for what actions, which actions need to be implemented annually to prepare the city and which flood resilience measures are needed and are most probable to be implemented in the city of Rethymno.

Following the workshop, a questionnaire was sent out to the participants of Rethymno's LAA, asking them to identify the types of floods that are affecting the city of Rethymno and their preferences in terms of the characteristics of measures that need to be implemented.

Stakeholders identified that a city may be protected using two approaches:

- The annual preparedness actions that prepare the city against the seasonal flooding period and may include actions of tidying up the city's streams and drainage system, informing local stakeholders and coordinating with the civil protection authority.
- The resilience measures that help to safeguard the city against floods by implementing new operational or structural measures which increase the resilience of the area.

While the first set, that of the preparedness actions, is usually already known by the local authorities and even described in the area's memorandum of actions for flood protection, the latter is the result of consultation with experts regarding the specific characteristics and the possible effects of the resilience measures to the area.

During the second stakeholder workshop [31], the participants were asked to identify the barriers for implementation and the operational and construction costs for all the actions and measures they had previously identified.

### 2.1.3. Evaluative Criteria and Outcomes

In this work, it was decided to use a qualitative indicator to evaluate the outcome of the flood risk management decision making that will estimate the performance of the area to manage flood risk. The severity of a flood is linked to the return period of the event and is commonly evaluated based on the damages that flood events cause. Additionally, the effectiveness of the implemented measures and actions is an input to the model that is identified using expert knowledge included within a knowledge base. The Preparing for Extreme and Rare Events in Coastal Regions (PEARL) intelligent knowledge-base (PEARL KB) of resilience strategies is an environment that allows end-users to navigate from their observed flood problem to a selection of possible options and interventions worth considering within an intuitive visual web interface assisting advanced interactivity. PEARL KB is available at pearl-kb.hydro.ntua.gr [32]. The performance of the area to manage flood risk is then estimated by decreasing the severity of the flood by adding the effectiveness of the implemented measures and actions. This qualitative indicator tries to overcome the necessity of estimating flood damages in order to assess the effectiveness of measures, mainly because the methodology aims at exploring scenarios so as to transfer knowledge to the decision makers and not to actually assess the outcome of actual flood risk management decisions.

### 2.2. PEARL ABM SAS

The Preparing for Extreme and Rare Events in Coastal Regions (PEARL) institutional ABM Simulating Authorities' decision making for the Selection of resilience strategies (PEARL ABM SAS) was created to support the exploration of alternative intervention options by the stakeholders of

flood risk management. PEARL ABM SAS simulates how authorities prepare against flood risk by implementing alternative intervention options under different socio-economic conditions and different flood event scenarios.

In this section the design concepts, the variables, and the processes of the PEARL ABM SAS model are presented using the ODD protocol [33], following recommendations of [34].

### 2.2.1. Design Concepts

Using the ODD terminology, PEARL ABM SAS's design concepts can be defined as follows:

1.  Emergence: Emergence in PEARL ABM SAS relates to the fact that the micro behaviour of each stakeholder agent results in the estimation of the macro behaviour of the city's flood risk management. This behaviour is then converted into city performance using qualitative metrics.
2.  Adaptation: Decision making agents adapt their behaviour depending on the available funding and the occurrence of flood events in the previous years.
3.  Fitness: The model was designed taking into account the beliefs and views of the risk flood management stakeholders. The model was first presented during the 2nd stakeholder workshop in Rethymno and it received positive feedback.
4.  Prediction: Decision making agents anticipate the increase of the performance of their city to protect, prevent, and prepare against flood events.
5.  Interaction: Decision making agents interact with stakeholder agents and if they cooperate actions and measures are implemented.
6.  Sensing: Decision making agents communicate with central authority agents to get information regarding the available to flood risk management funding. Additionally, decision making agents get information regarding the specific characteristics (cost, cooperating parties etc.) of measures and actions under consideration.
7.  Stochasticity: The cooperation or not between the authorities and the stakeholders in some cases is randomly selected giving a stochastic nature to the model's behaviour.

### 2.2.2. State Variables and Scales

The low-level parameters of the agents and the ABM [33] and their descriptions are given in Appendix A Table A1 categorised per decision procedure where appropriate.

### 2.2.3. Process Overview and Scheduling

The PEARL ABM SAS simulates the authorities' decision-making process for the selection of resilience strategies and assesses the performance of the case area under different socio-economic and flood events scenarios. The local authorities' agents have preferences and interests regarding the city's flood protection. It is assumed that the decision-makers' behaviour will be influenced by the available funding sources, their preferences and the political will generally for urban water management system and defined by specific rules of conduct. The intelligent agents will interact to implement actions for flood preparedness and will decide on the implementation of new measures in response to external pressures such as flooding events. Additionally, the interaction of intelligent agents will simulate the cooperation between the different services, and follow basic principles of game theory (such as the prisoner's dilemma, etc.), which allows for greater realism in modelling collaborations, since it may reflect a lack of cooperation phenomena even when the cooperative behaviour appears more beneficial for all parties [35].

The agents follow specific rules that allow them to get information regarding the available funding and the characteristics of the resilience strategies, interact with the stakeholders to prepare the city for flooding events and implement actions, decide to implement new flood resilience measures, inspect existing flood resilience measures, and maintain them (Figure 3).

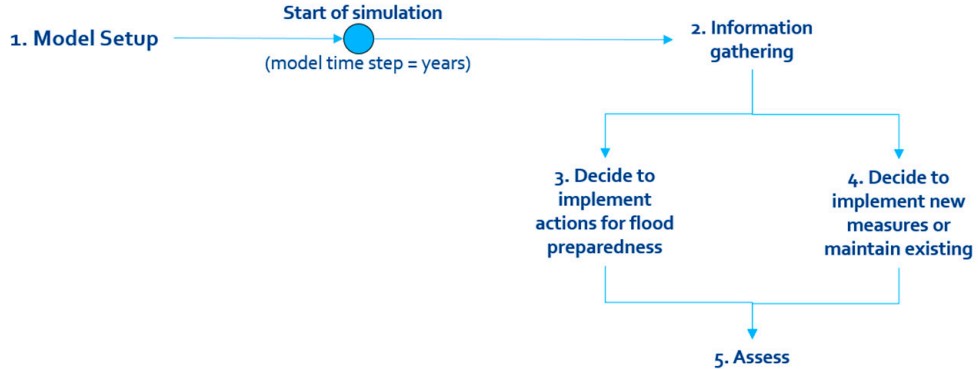

**Figure 3.** Core procedure of the PEARL ABM SAS.

As depicted in Figure 3 the PEARL ABM SAS core procedure is as follow:

1.  Model setup: The model is initialised by setting up the environment and the specific characteristics of the agents. The model has an annual step and the simulation has been defined to have a span of ten years, to allow the experimentation with different scenarios of flooding occurrence and socio-economic conditions.

2.  Information gathering: The local authorities' decision-making agents get information regarding the available funding, the characteristics of flood measures (link with PEARL KB) available for assessment and the weights of importance of each flood resilience measures' characteristic. Establishing communication between local and central authorities is linked to the variable "bilateral communication" which is set based on expert knowledge based on the level of cooperation between the authorities (Figure 4).

3.  Decide to implement actions for flood preparedness: Agents cooperate with the stakeholders to implement several flood preparedness actions. The actions, their priorities, and their positive effects are identified outside the ABM by expert knowledge and may incorporate the findings of LAAs and stakeholder participation processes. The cooperation of the agents is based on a one time prisoner dilemma game that tries to simulate the cooperation procedures of the authorities. The prisoner's dilemma rules are given in the following table. They were selected so as to depict both incentives of both cooperation or not. If the agents decide to cooperate, then actions are implemented only if available funds exist (Figure 5).

4.  Decide to implement new measures or maintain existing: Decision making agents cooperate with the Water Utility to implement or maintain flood resilience measures (Figure 6). The list of measures and their specific characteristics like the construction and operation cost, are identified outside the ABM by expert knowledge. The measures are characterized in the PEARL Knowledge Base to which the decision-making agents have access. Local authorities prioritise the implementation or maintenance of measures by performing a multi criteria assessment of all the flood resilience measures under investigation using the following rule:

$$Sm = \sum_{i=0}^{10} W_i \times C_i$$

where $Sm$ is the total score of a measure, $W_i$ is the weight of a measure's characteristics based on the decision making authority's preferences, and $C_i$ is the measure's characteristic.

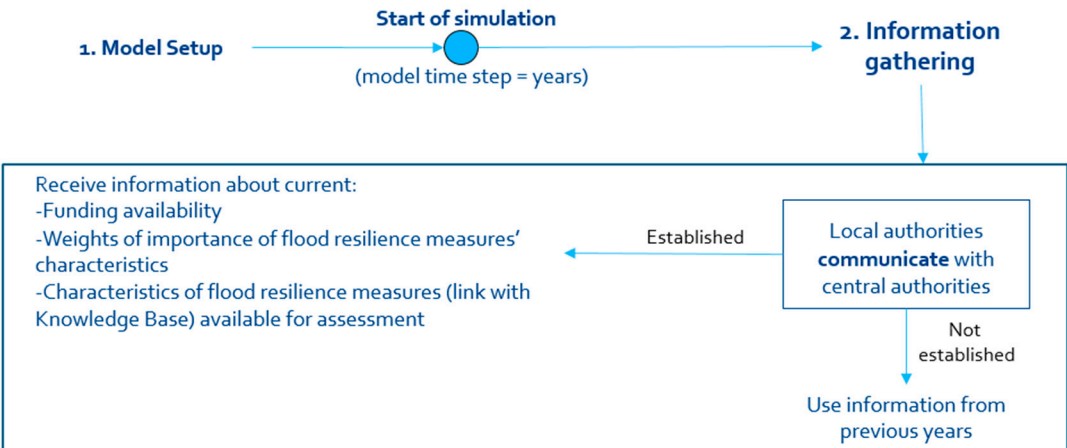

**Figure 4.** Procedure for information gathering.

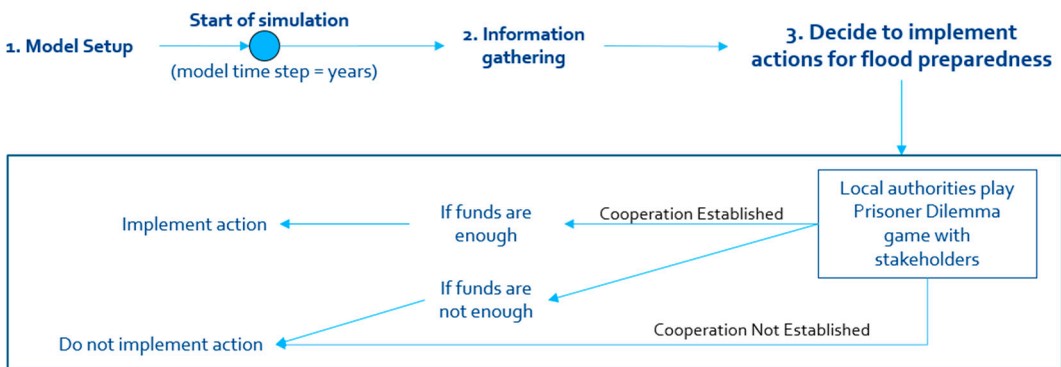

**Figure 5.** Procedure for implementing actions for flood preparedness.

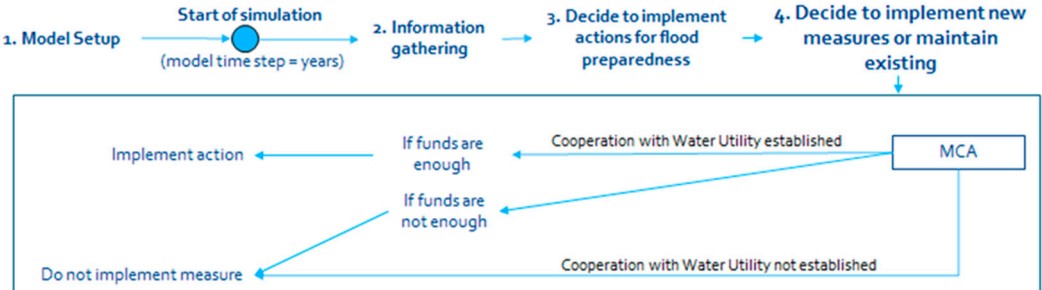

**Figure 6.** Procedure for implementing new measures or maintain existing.

The weights of a measure's characteristics are identified outside the ABM by expert knowledge. In order to acquire the weights, firstly experts perform a pairwise comparison of how important each characteristic of the measure is. Then, this "importance score" is transformed into weights using the analytical hierarchy process.

5.    Assess: the performance of the area in being prepared and protected by the flooding events that affect the area. It is assumed that a city that implements all proposed actions is prepared and protected from average events (in this experimental design, medium severity events of both pluvial and coastal type). The performance of the area is relevant to the positive impacts of the implemented flood preparedness actions and flood resilience measures.

2.2.4. User Interface

The PEARL ABM SAS is accessible via the PEARL Toolbox [17] (accessible from http://pearl-kb.hydro.ntua.gr/tb/). By clicking on the ABM PEARL SAS button from the main page of the PEAL Toolbox, the user is navigated to the homepage of the PEARL ABM SAS (Figure 7).

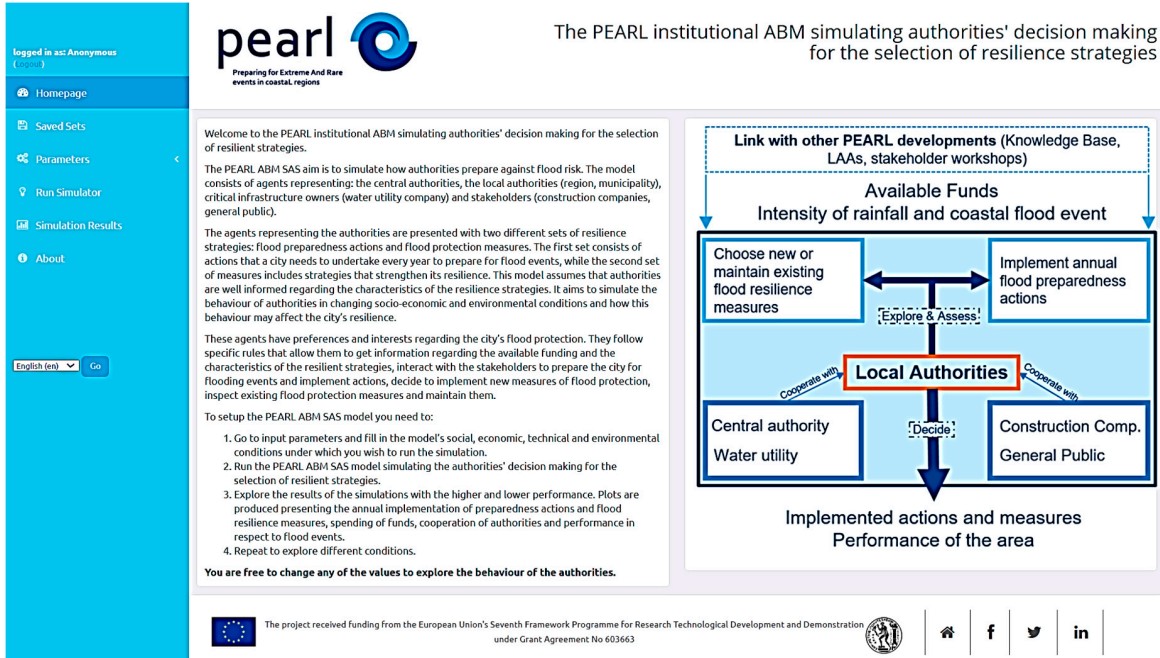

**Figure 7.** Homepage of the PEARL institutional ABM simulating authorities' decision making for the selection of resilience strategies.

This web-based interface was designed and developed aiming to make the developed tool more user-friendly and available to decision makers for use. The PEARL ABM SAS model was created using NetLogo [36]. The interface is linked to a webserver that operates the NetLogo model and saves the results and presents them to the user, as depicted in Figure 8.

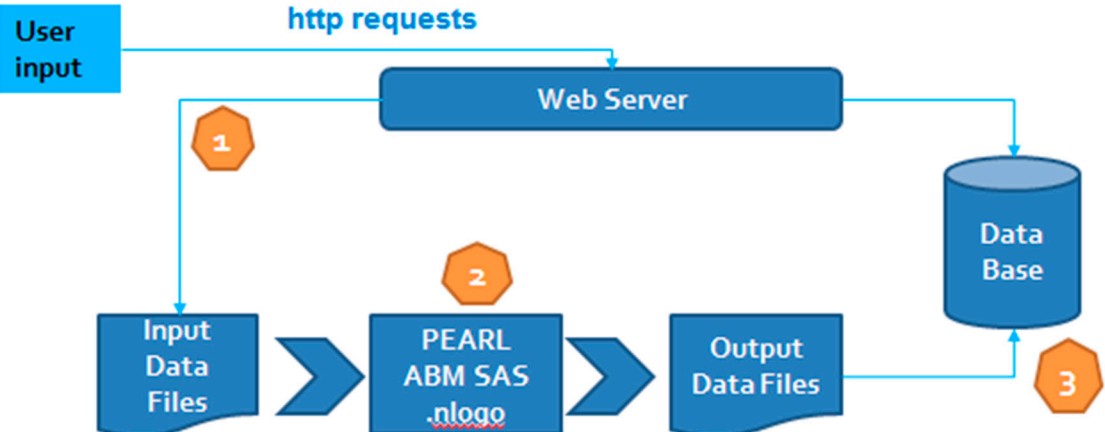

**Figure 8.** Online interface link to NetLogo, database and web server. 1. Web Server creates Input data Figure 2. ABM parses input data file, simulates the decision making and creates output data file with results (process is triggered by Web Server) and 3. With the help of the Web Server results are inserted in the database, ready to be shown to the user.

In order to start a simulation, the user has to specify the input parameters first by clicking on Input parameters from the ABM PEARL SAS menu. Each sub-menu opens a web page where the user is able to parameterize the model. See Figure 9 for an example of the user interface for setting up the flood preparedness actions.

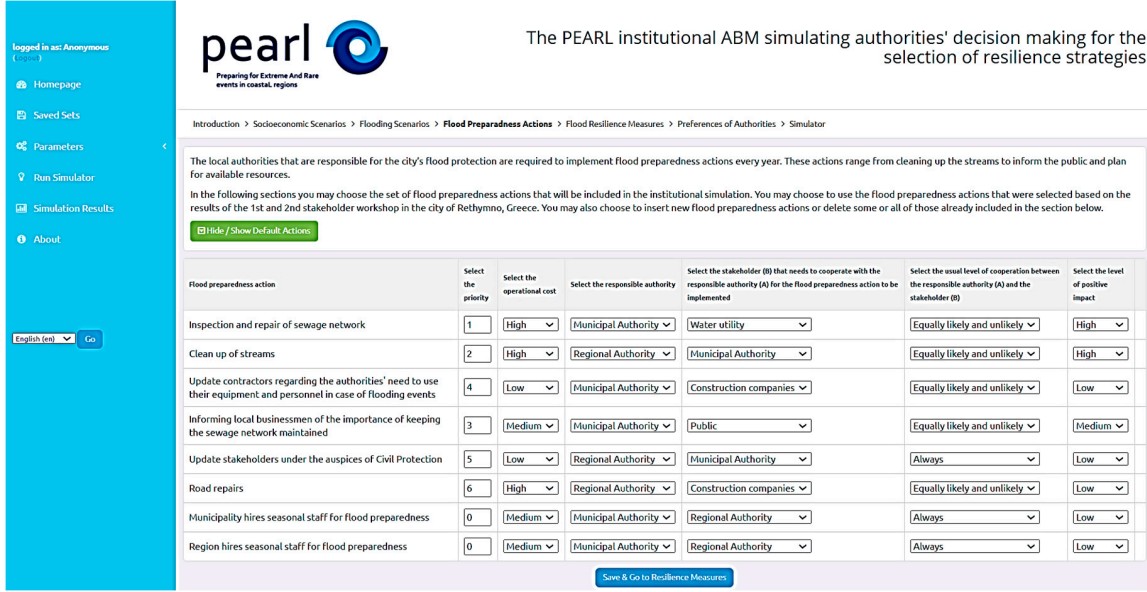

**Figure 9.** Choose flood preparedness actions and their characteristics.

Additionally, the importance of the different characteristics of the flood resilience measures is assessed based on a pairwise comparison that is included in the menu preference of authorities (Figure 10). The interface allows the user to give different preferences for different responsible authorities (municipality and prefecture), thus enabling to model the preferences of different decision making authorities and how these affect their decisions.

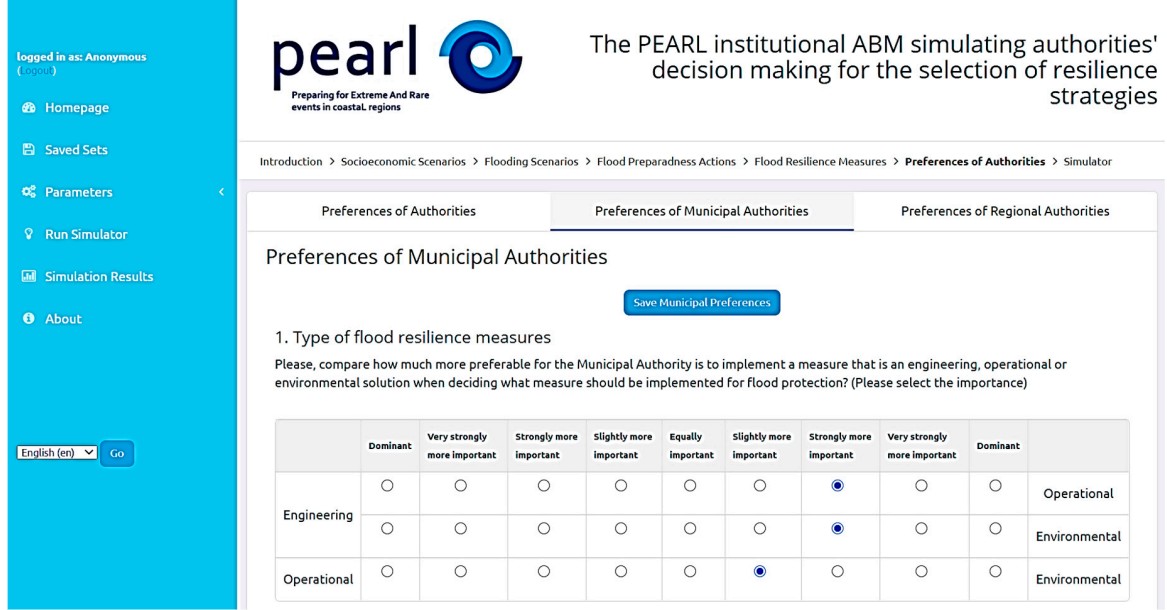

**Figure 10.** Preferences of authorities.

Using the online interface, the user is able to run the PEARL ABM SAS without having any prior knowledge on how to use ABMs or NetLogo. The user is able to run the simulation using the Run Simulator menu and obtain results through the Simulation results menu. The user is also able to save and retrieve previous runs by importing saved sets of parameters using the menu Saved sets.

By the end of the PEARL ABM SAS simulation, the simulation output is automatically produced. The last output is also saved together with the saved parameter set to be able to trace back the results of different saved parameter sets. The simulation output presents two selected simulation results, those with the higher and lowest performance in respect to the preparation and protection of the city against the flooding scenarios given the selected social, economic, technical and environmental conditions. First, the implementation of preparedness actions is given, where the implemented preparedness actions are shown during the 10 years simulation. If an action is implemented during a year, the box is coloured green and if it is not the box is colored grey.

## 3. Results

### 3.1. Setup of PEARL ABM SAS for the Rethymno Case Study

The application of the institutional analysis framework to the flood risk management of the city of Rethymno and the results of the questionnaire and the stakeholder workshops were used to design the PEARL ABM SAS. The participants of the LAA workshops in Rethymno were involved in the entire process of the PEAL ABM SAS design and development [30,31]. The workshops' participants were asked to define the socio-economic characteristics of Rethymno [30]. Their recommendations were used to setup the PEARL ABM SAS for exploring the behaviour of Rethymno case study. The following tables present the selection of characteristics to setup the PEARL ABM SAS to depict the Rethymno case study. The socio-economic conditions specifically are being described in Table 2.

**Table 2.** Socio-economic conditions.

| | |
|---|---|
| **Funding** | **Insufficient Funds. The Authorities Have Some, Inadequate, Funding for Preparing and Protecting the City Against Flood Events.** |
| **Local Political Will** | Small interest. A city must be prepared for the most common non-severe flood events. |
| **Lateral Communication** | Yes. Local authorities and central authorities are able to communicate and exchange information regarding available funding, political interests and preferences regarding the flood protection and preparedness. |
| **Corruption** | No. Decisions are made and funds are spent implementing the selected flood protection and preparedness strategies. |

The flood preparedness actions which are under consideration for the specific case study are the following, and their corresponding defined parameters are presented in Table 3.

A. Inspection and repair of sewage network
B. Clean up of streams
C. Informing local businessmen of the importance of keeping the sewage network maintained
D. Maintain existing flood protection measures
E. Update contractors regarding the authorities' need to use their equipment and personnel in case of flooding events
F. Update stakeholders under the auspices of civil protection
G. Road repairs

**Table 3.** Parameters defined related to flood preparedness actions under consideration.

|  | Priority | Operational Cost | Responsible Authority (A) | Stakeholder (B) | Usual Level of Cooperation between (A) and (B) | Level of Positive Impact |
|---|---|---|---|---|---|---|
| A | 1 | Medium | Municipal Authority | Water Utility | Equally likely and unlikely | High |
| B | 2 | Low | Regional Authority | Municipal Authority | Always | High |
| C | 3 | Low | Municipal Authority | Public | Equally likely and unlikely | High |
| D | 4 | Low | Municipal Authority | Regional Authority | Equally likely and unlikely | Medium |
| E | 5 | Low | Municipal Authority | Construction companies | Equally likely and unlikely | Low |
| F | 6 | Low | Regional Authority | Municipal Authority | Always | Low |
| G | 7 | High | Regional Authority | Construction companies | Equally likely and unlikely | Low |

In terms of flood resilience strategies which are under consideration by the stakeholders of Rethymno, the following list summarizes them, and Table 4 presents their corresponding parameters.

1. Beach nourishment
2. Breakwater
3. Floodwall
4. Increased capacity of sewer/drainage system
5. Check valve
6. Flood detention reservoir
7. Evacuation plan
8. Flood forecasting and early warning
9. Land use plan/spatial planning
10. Public awareness information education and communication
11. Flood insurance

**Table 4.** Parameters defined related to flood strategies under consideration.

|  | Construction Cost | Time in Months Required until the Measure is Operational | Operational Cost for Each Measure | Responsible Authority |
|---|---|---|---|---|
| 1 | Low | 3 | Medium | Municipality |
| 2 | High | 36 | Low | Region |
| 3 | Medium | 36 | Low | Municipality |
| 4 | Medium | 6 | Medium | Municipality |
| 5 | Medium | 6 | Low | Municipality |
| 6 | High | 48 | Medium | Region |
| 7 | Low | 24 | Low | Municipality |
| 8 | Medium | 36 | Medium | Municipality |
| 9 | Medium | 48 | Medium | Municipality |
| 10 | Low | 3 | Low | Municipality |
| 11 | Medium | 24 | Medium | Municipality |

The stakeholders used a questionnaire with pairwise comparison of each one of the measures' characteristics. Their answers were used to estimate the authorities' weights of the measures' characteristics and are presented in the following table (Table 5).

**Table 5.** Weight of the characteristics of flood resilience measures stored in the KB.

| Type | Weight | Timescale | Weight |
|---|---|---|---|
| Engineering | 0.09 | Long term | 0.75 |
| Environmental | 0.30 | Medium term | 0.12 |
| Operational | 0.61 | Short term | 0.13 |
| **Approach** | **Weight** | **Target** | **Weight** |
| Protection approach | 0.75 | Mitigation | 0.75 |
| Accommodation approach | 0.12 | Adaptation | 0.25 |
| Retreat approach | 0.13 | | |
| **Problem** | **Weights** | **Problem (Continue)** | **Weight** |
| Water retention or detention | 0.29 | Financial preparedness | 0.03 |
| Rivers' Capacity Enhancement | 0.14 | Flood preparedness | 0.03 |
| Coastal management | 0.10 | Emergency response | 0.02 |
| Conventional urban drainage | 0.07 | Green measures | 0.02 |
| Source control | 0.06 | Blue measures | 0.02 |
| Infiltration and buffering technique | 0.05 | Building flood proofing | 0.02 |
| Conveyance & Storage Structure | 0.04 | Governance and Policies | 0.02 |
| Information, Education & Communication | 0.04 | Assessment and Evaluation | 0.02 |
| Land use control | 0.03 | Recovery | 0.02 |
| **Land Use** | **Weight** | **Flood Type** | **Weight** |
| Urban | 0.34 | Coastal | 0.38 |
| Suburban | 0.15 | Fluvial | 0.03 |
| Rural | 0.06 | Pluvial | 0.35 |
| Coastal | 0.37 | Groundwater | 0.04 |
| Industrial | 0.06 | Drain & Sewer | 0.19 |
| Park | 0.02 | | |
| **Spatial Scale** | **Weight** | | |
| River Basin | 0.04 | | |
| City | 0.42 | | |
| Neighborhood | 0.30 | | |
| Street | 0.17 | | |
| Building | 0.07 | | |
| **Operational Cost** | **Weight** | **Construction Cost** | **Weight** |
| OC High | 0.09 | CC High | 0.53 |
| OC Medium | 0.38 | CC Medium | 0.30 |
| OC Low | 0.53 | CC Low | 0.17 |

### 3.2. Results from the Collaborative Analysis of the PEARL ABM SAS in Rethymno

At the final LAA workshop (29 September 2017 in Rethymno, Crete) the workshop's participants were introduced to the final version of the PEARL ABM SAS, the user interface, and the linking with the PEARL knowledge base, which they were already acquainted with. During this final workshop the stakeholders were able to use the tool and understand the usability of the PEARL ABM SAS and the tool's results. Upon the completion of the exercise the participants and the researchers were involved in a long discussion regarding the usability of the tool by policy makers. The results that the participants saw were those of the user interface that present the best and worst performance of 10 simulation runs.

Several flooding scenarios were used to explore the behavior of flood risk management in Rethymno. In the PEARL ABM SAS, the implementation or not of an action is related to both the flooding events of the previous year and the selected socio-economic conditions of the area. Stakeholders identified that authorities implement an action if funding is available and if they cooperate

with other stakeholders without the interference of the decision-making process by corruption and/or lack of lateral communication between authorities. Additionally, extra measures for flood protection are usually implemented only if funds are available and authorities cooperate. The above signifies how little effect previous flooding events have on the decisions of authorities, especially when political will is weak.

The workshop's participants validated that the results of the PEARL ABM SAS are in line with the actual reality and that the actions that are usually implemented to prepare for floods in Rethymno are the same with those that the agent-authorities implement. Another important part of the model that was validated during the workshop was the effect of the collaboration between the different authorities and the importance of political will.

In order to examine the results of the PEARL ABM SAS here, it was decided to collect the results of 100 simulation runs, in order to decrease model outliers (see [6]), with the same initial setup and present them in an implementation heatmap. The heatmap shows how many times an option (action or measure) is implemented for each simulation year summing up the results of the 100 simulation runs.

Rethymno Flood Scenarios

Figure 11 presents the heatmap of the implementation of actions and of measures for three different flood scenarios, a no flood scenario where no floods are recorded in 10 years, an annual medium flood scenario, where floods of low intensity (T < 100 year-flood) are reported every year and an annual high flood scenario, where floods of severe intensity (T > 100 year-flood) are reported every year.

It is evident that, the increase of flood intensity does not significantly affect the decisions of authorities, even though there is a conceptual link between flooding events and political will. Additionally, in terms of specific actions, those with high cost, low impact, and uncertain collaboration between the authorities and the stakeholders tend to be the least implemented actions. This verifies that PEARL ABM SAS behaves as expected.

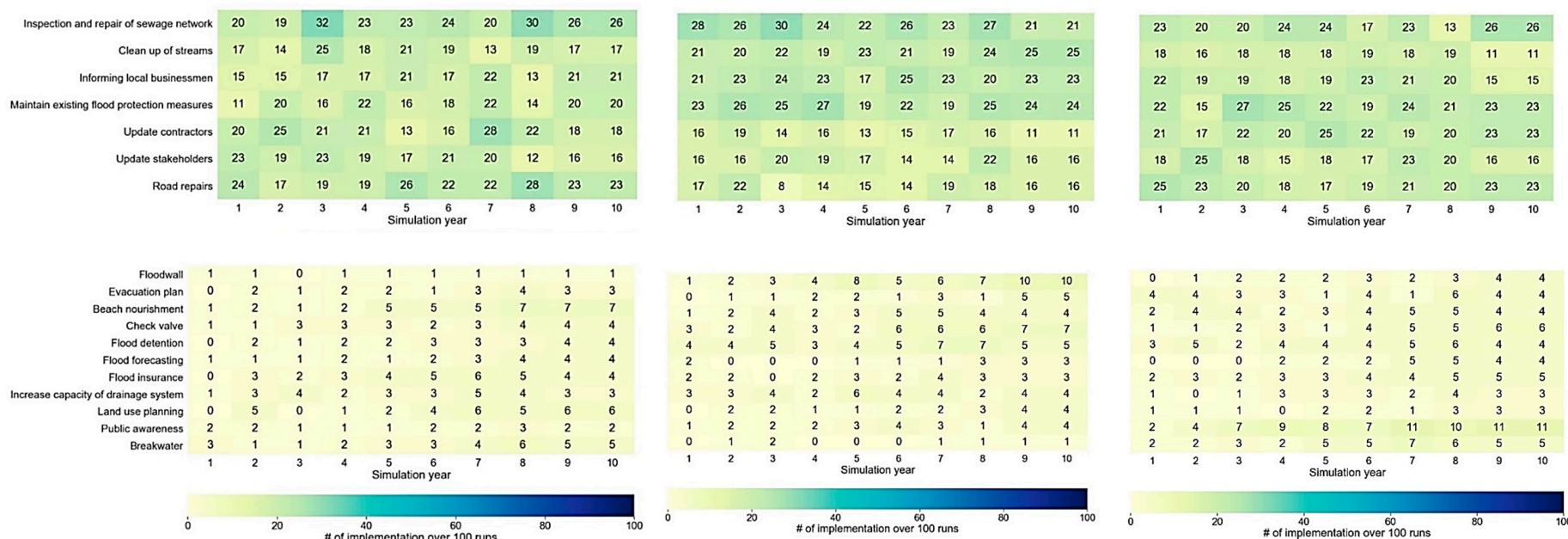

**Figure 11.** Heatmap of implemented actions (**up**) and measures (**down**) per simulation year for no (**left**), low (**middle**) and high (**right**) flood events annually with insufficient funding and small interest from authorities.

The local stakeholders defined funding, cooperation and political will as the main drivers of flood risk management in Rethymno. Figure 12 presents the effects of funding increase, political will increase and stakeholder better cooperation in the average implementation rate (average number of times an option is implemented within the 10-year simulation period and for all the 100 simulation runs (max = 100)). It is noted that the effect of political will and cooperation are not significant, with the evident increase or decrease being mainly attributed to the model's uncertainty. Nevertheless, the effect of funding is undeniably present with an increase of more than 100% of the implementation times of the actions. Figure 13 presents the heatmap of the implementation of preparedness actions with an annual medium flood scenario, where floods of medium intensity (T < 100 year-flood) and enough funding for flood risk management. Comparing with the middle upper plot of Figure 11, the effects of the increase of funding are manifested.

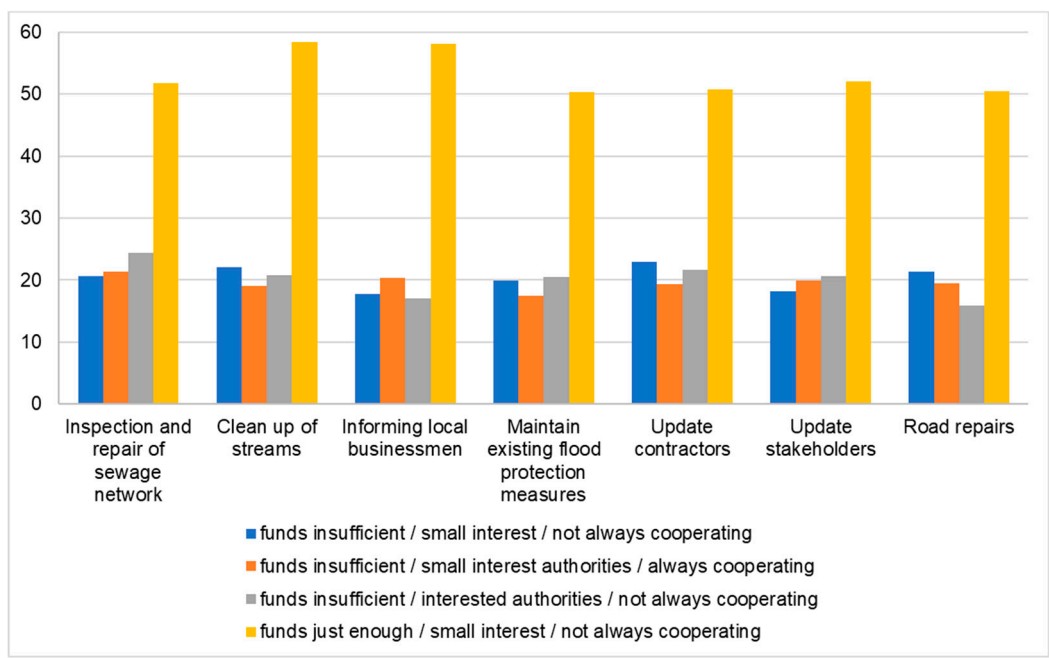

**Figure 12.** Comparison of average times of implementation of flood preparedness actions for different socio-economic scenarios.

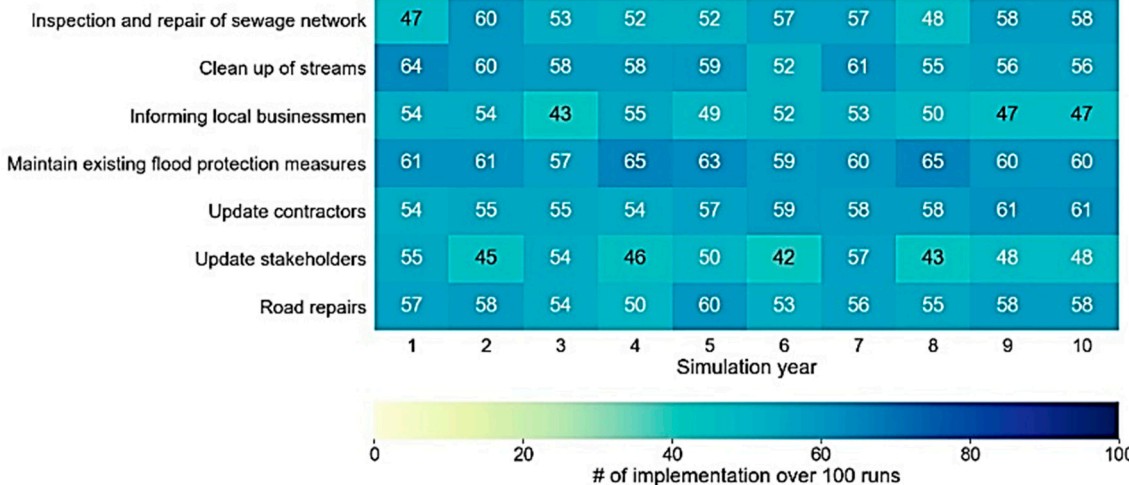

**Figure 13.** Heatmap of how many times (#) actions have been implemented per simulation year with annual low flood severity (T < 100 year-flood) and just enough funding.

## 4. Discussion

PEARL ABM SAS simulates the authorities' decision-making process for the selection of resilience strategies and assesses the performance of the case area under different socio-economic and flood events scenarios. The model was designed using the institutional analysis framework of the Nobelist Ellinor Ostrom.

PEARL ABM SAS, through its easy to use user interface, provides a useful and a tangible way for authorities to examine and explore factors affecting the actual implementation of a flood protection measure, after the decision that a measure is needed. Such factors are related to the available funding, the level of authorities' cooperation, the political will regarding flood risk management etc.

The validation of the PEARL ABM SAS was performed by the stakeholders of flood risk management in Rethymno. Additionally, the workshop participants used it to explore the effect of different flood scenarios on the agents' decisions. The stakeholders identified a leverage point in the decision-making process, which is that of the cooperation between the local authorities and the stakeholders. While this was a known difficulty for the flood risk management of the city the use of the PEARL ABM SAS enabled the stakeholders to experiment with different collaboration approaches and discuss between them their effects on the protection of the city.

The produced computational experiments were therefore able to inform authorities about the effects of their institutional behaviour. This exploration and the experimentation with these drivers (i.e., availability of funding, cooperation levels etc.) started discussions between the decision makers regarding ways to increase cooperation and plan ahead. For example, during the workshop, the stakeholders identified timing as one of the main barriers for collaboration, which could possibly be solved by using better project management tools, online collaboration tools, and allowing a wider window for communication. For example, specifically for preparedness action "B. Clean up of steams", it was discussed how the action is linked with information campaigns and school participation, which most of the time shifts its period of implementation from August–September to May–June, with no effects whatsoever on the flood events which usually occur in late autumn and early winter.

PEARL ABM SAS could be used in other areas as well, since the setup (see Section 3.1) can be altered easily using its user interface, as a tool that enables stakeholders to experiment with their flood risk management decision making process and find leverage points that could increase the performance of the area against floods.

**Author Contributions:** Conceptualization, I.K. and C.M.; data curation, I.K. and A.L.; formal analysis, I.K.; funding acquisition, C.M.; investigation, I.K. and A.L.; methodology, I.K.; project administration, C.M.; resources, I.K. and A.L.; software, I.K. and C.P.; supervision, C.M.; validation, I.K. and A.L.; visualization, I.K. and C.P.; writing—original draft, I.K.; writing—review & editing, I.K., A.L. and C.M. All authors have read and agreed to the published version of the manuscript.

**Funding:** The research leading to these results has received funding from the European Union Seventh Framework Programme (FP7/2007–2013) under Grant agreement No. 603663 for the research project PEARL (Preparing for Extreme and Rare events in coastaL regions). The research and its conclusions reflect only the views of the authors and the European Union is not liable for any use that may be made of the information contained herein.

**Acknowledgments:** The authors would like to thank Patricia Gourgoura who assisted in the design, development and implementation of the stakeholder workshops.

**Conflicts of Interest:** The authors declare no conflict of interest.

# Appendix A

**Table A1.** Low level parameters of PEARL ABM SAS.

| PEARL ABM SAS Components | Parameters | Brief Description |
|---|---|---|
| Scenarios flooding | Time series of 10 values of annual occurrence of floods and their intensity. | Scenario values correspond to: No flood events = 0, Medium flood events (T < 100 year-flood) = 1, Extreme flood events (T > 100 year-flood = 2. |
| Measures | Number of measures (#) | Total number of measures under investigation. |
| | Responsible authority | Select the authority responsible for the decision: Municipality = 1 or Region = 2 |
| | Impact of the measure if applied in flood protection. | Qualitative assessment of the impact, taken from the PEARL Knowledge Base: Low = 1, Medium = 2, High = 3. |
| | Measure's characteristics from PEARL Knowledge Base | 1. Type: Engineering, Environmental, Operational 2. Timescale: Long, Medium or Short term 3. Approach: Protection, Accommodation, Retreat approach 4. Problem: Water retention or detention, Rivers'/Waters' Capacity Enhancement, Coastal management, Conventional urban drainage, Source control, Infiltration and buffering technique, Conveyance & Storage Structure, Information, Education & Communication, Land use control, Financial preparedness, Flood preparedness, Emergency response, Green measures, Blue measures, Building flood proofing, Governance and Policies, Assessment and Evaluation, Recovery 5. Target: Mitigation, Adaptation 6. Land use: Urban, Suburban, Rural, Coastal, Industrial, Park 7. Flood type: Coastal, Fluvial, Pluvial, Groundwater, Drain & Sewer 8. Spatial scale: River Basin, City, Neighbourhood, Street, Building 9. Operational cost: High, Medium, Low 10. Construction cost: High, Medium, Low |
| Actions | Number of actions (#) | Total number of actions under investigation. |
| | Name of actions | Name of action given by the user. |
| | Operational cost | Qualitative assessment of cost: Low = 1, Medium = 2, High = 3 |
| | Responsible authority (municipality or region) | Select the authority responsible for the decision: Municipality = 1 or Region = 2 |
| | Priority | Prioritise actions based on the user's opinion. |
| | Positive effect of action | Qualitative assessment of the presumed impact of the action: Low = 1, Medium = 2, High = 3. |
| | Implementation process | Define which authority is responsible and with which stakeholder needs to cooperate in order for the action to be implemented. Create links between authority agents. |
| Authorities-agents | Preferences | User's weights of importance for the measure's characteristics. |
| | Authorities' interest to flood protection and preparedness | Qualitative assessment of the interest: Low = 1, Medium = 2, High = 3. |
| | Available funds | Qualitative assessment of the availability of funds: Low = 1, Medium = 2, High = 3. |
| | Bilateral communication | Qualitative assessment of the communication between local and central authorities: Yes = 1, No = 2 |
| | Corruption | Qualitative assessment of the existence of corruption: Yes = 1, No = 2 |
| Stakeholders-agents | Level of cooperation | Qualitative assessment of the cooperation between the stakeholders and the authorities: Never = 0, Always = 1, Equally likely and unlikely = 2 |

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
