# Peer review of "Investigating Decision Mechanisms of Statutory Stakeholders in Flood Risk Strategy Formation: A Computational Experiments Approach"

_water, doi:10.3390/w12102716_

Round 1
Reviewer 1 Report
The topic of this paper is for sure interesting because it shows a great solution for the administrations to start facing the challenges of urban context prone to flooding risk.
Along with small revisions of the English translation, I think that the introduction should contain some additional information. It is necessary to explain in a more detailed way the state of art to understand the issue (more citations are also needed) and to underline the objective of this research work.
I suggest also to improve the distinction between the different sections of the paper, especially in the “material and methods” chapter. The “material (or similar)” paragraph should collect all the information related to the instruments used for this analysis (i.e. previous models).
I suggest also a revision of the tables, especially Table 2 that is too long and not very well expressed. Some information, probably, can be also written in the text.
Reviewer 2 Report
Evaluation of the manuscript entitled:
Investigating decision mechanisms of statutory stakeholders in flood risk strategy formation: a computational experiments approach.
By Ifigeneia Koutiva, Archontia Lykou, Chris Pantazis, Christos Makropoulos
This very interesting study presents a new approach to explore the effects that authorities’ behaviour has to the decisions for preparing and protecting a city against floods. For this, a tool developed to enable the experimentation with the decision-making process of authorities responsible for a city’s flood risk management.
The tool has a user-friendly interface enabling the end-users to explore the drivers of decision-making processes under different conditions. The tool used as a case study the responsible authorities for flood protection in the city of Rethymno in the island of Crete, Greece.
The topic is very interesting and the paper is very well written. In general the methodology is well explained and the results/discussion is covering all aspects of the study. The subject is in the scope of the journal.